# The importance of oxytocin neurons in the supraoptic nucleus for breastfeeding in mice

**Mitsue Hagihara⦿, Kazunari Miyamichi ⓘ \*, Kengo Inada ⓘ \*⦿**

RIKEN Center for Biosystems Dynamics Research, Chuo-ku, Kobe, Hyogo, Japan

⦿ These authors contributed equally to this work.
\* k.inada.repository@gmail.com (KI); kazunari.miyamichi@riken.jp (KM)

**Data Availability Statement:** All relevant data are within the paper.

**Funding:** K.I. was supported by the RIKEN Special Postdoctoral Researchers Program (https://www.

## Abstract

The hormone oxytocin, secreted from oxytocin neurons in the paraventricular (PVH) and supraoptic (SO) hypothalamic nuclei, promotes parturition, milk ejection, and maternal caregiving behaviors. Previous experiments with whole-body *oxytocin* knockout mice showed that milk ejection was the unequivocal function of oxytocin, whereas parturition and maternal behaviors were less dependent on oxytocin. Whole-body knockout, however, could induce the enhancement of expression of related gene(s), a phenomenon called genetic compensation, which may hide the actual functions of oxytocin. In addition, the relative contributions of oxytocin neurons in the PVH and SO have not been well documented. Here, we show that females with conditional knockout of *oxytocin gene* in both the PVH and SO undergo grossly normal parturition and maternal caregiving behaviors, while dams with a smaller number of remaining *oxytocin*-expressing neurons exhibit severe impairments in breastfeeding, leading to the death of their pups within 24 hours after birth. We also found that the growth of pups is normal even under *oxytocin* conditional knockout in PVH and SO as long as pups survive the next day of delivery, suggesting that the reduced oxytocin release affects the onset of lactation most severely. These phenotypes are largely recapitulated by SO-specific *oxytocin* conditional knockout, indicating the unequivocal role of oxytocin neurons in the SO in successful breastfeeding. Given that oxytocin neurons not only secrete oxytocin but also non-oxytocin neurotransmitters or neuropeptides, we further performed cell ablation of oxytocin neurons in the PVH and SO. We found that cell ablation of oxytocin neurons leads to no additional abnormalities over the *oxytocin* conditional knockout, suggesting that non-oxytocin ligands expressed by oxytocin neurons have negligible functions on the responses measured in this study. Collectively, our findings confirm the dispensability of oxytocin for parturition or maternal behaviors, as well as the importance of SO-derived oxytocin for breastfeeding.

## Introduction

Oxytocin (OT) is a nonapeptide hormone produced by OT neurons in the paraventricular (PVH) and supraoptic (SO) hypothalamic nuclei. Recent studies have reported that OT plays

riken.jp/en/careers/programs/spdr/), a grant from the Kao Foundation for Arts and Sciences (https://www.kao-foundation.or.jp/english.html), and Japan society promotion science KAKENHI (19J00403 and 19K16303) (https://www.jsps.go.jp/english/) K.M. was supported by Japan society promotion science KAKENHI (20K20589 and 21H02587) (https://www.jsps.go.jp/english/). The funders had no role in study design, data collection and analysis, decision to publish, or preparation of the manuscript. The funder (RIKEN Special Postdoctoral Researchers Program) provided support in the form of salaries for the last author [KI], but did not have any additional role in the study design, data collection and analysis, decision to publish, or preparation of the manuscript. The specific roles of the author are articulated in the 'author contributions' section.

**Competing interests:** The authors have declared that no competing interests exist.

important roles in sexual, maternal, and social behaviors [1–3], in addition to the functions documented in classical studies such as the induction of labor and milk ejection. However, studies on whole-body *OT* knockout (KO) mice have shown that milk ejection is a specific and essential function of OT, but dispensable for parturition [4–6]. Similarly, the expression of maternal caregiving behaviors does not require OT, given that the performance of parental behaviors by *OT* KO dams is largely similar to that by controls [5, 7], except in food-limited stressful environments [8]. Despite the clear consistencies across studies, the significance of OT signaling in the regulation of labor, milk ejection, and parental behaviors remains unclear, given that the phenotypes of a whole-body KO might be genetically compensated by the upregulation of related gene(s) [9, 10]. For example, a recent study analyzing a different function of OT neurons, weight homeostasis [11–13], reported that PVH-specific *OT* conditional KO (cKO) mice showed a hyperphagic obesity phenotype that was not apparent in the whole-body *OT* KO [14]. In addition, although OT neurons can release not only OT, but also other neurotransmitters or neuropeptides, such as glutamate [15], to our knowledge, the functional roles of such non-OT ligands in the regulation of labor, milk ejection, and parental behaviors have not been described.

Here, we show the relevance of OT secretion on labor, milk ejection, and parental behaviors using an *OT* cKO mouse line described previously [16]. Our approach offers a better temporal resolution, which allows us to avoid the influence of possible developmental or genetic compensation [9, 10]. We also test the relative contributions of OT secretion from the PVH and SO nuclei to maternal physiology and behaviors by restricting the manipulation to a single hypothalamic nucleus. This improved spatial resolution may reveal distinct functions of OT neurons in the PVH and SO, which show distinct input–output organizations [17]. Furthermore, we compare the phenotype of *OT* cKO with that observed in an OT neuron-specific cell ablation experiment, in which we test an additional function of non-OT ligands expressed in the OT neurons. Taken together, we aim to improve the resolution of loss-of-function studies of OT ligand and OT neurons in terms of maternal physiology and behaviors.

## Results

### *OT* cKO mothers in both PVH and SO fail in raising pups

Previous studies with whole-body *OT* KO female mice showed that fertility and pregnancy were unaffected [4–6], whereas parturition was either normal or delayed depending on the genetic background [5]. To evaluate such phenotypes in *OT* cKO females, we prepared $OT^{flox/flox}$ mice described previously [16]. In this line, Cre expression deletes floxed exon 1 of the *OT* gene, resulting in the loss of transcription of *OT* mRNA. We crossed *OT* KO ($OT^{-/-}$) and $OT^{flox/flox}$ mice and prepared $OT^{flox/-}$ mice [16]. Each $OT^{flox/-}$ female mouse was injected with an adeno-associated virus (AAV) driving Cre into both the PVH and SO (Fig 1A and 1B). Five weeks or longer after the injection, which was sufficient to reduce significantly the expression of *OT* mRNA in the PVH and SO (Fig 1C and 1D) [14], each female was crossed with a wild-type male (Fig 1B; Materials and Methods). We found that the durations of both pregnancy and pup delivery were indistinguishable between dams that had received vehicle (control) and *AAV-Cre* (+Cre) (Fig 1E–1G). However, as described previously [4–6], we found that in 54% (= 7/13) of the +Cre mothers' cages, all pups were dead in the next day of birth (postpartum day 1 [PPD 1]) without any sign of infanticide, even though they were alive at PPD 0 (Materials and Methods). This phenotype was not observed in the cages of vehicle-injected mothers (0/9). The fraction of surviving pups after PPD 1 was largely binomial: nearly 100% of pups survived in the cages of "success" dams, whereas 0% survived in the cages of "failure" dams (Fig 1H). By visualizing the *OT* mRNA and counting *OT* mRNA-positive (*OT*+) neurons, we

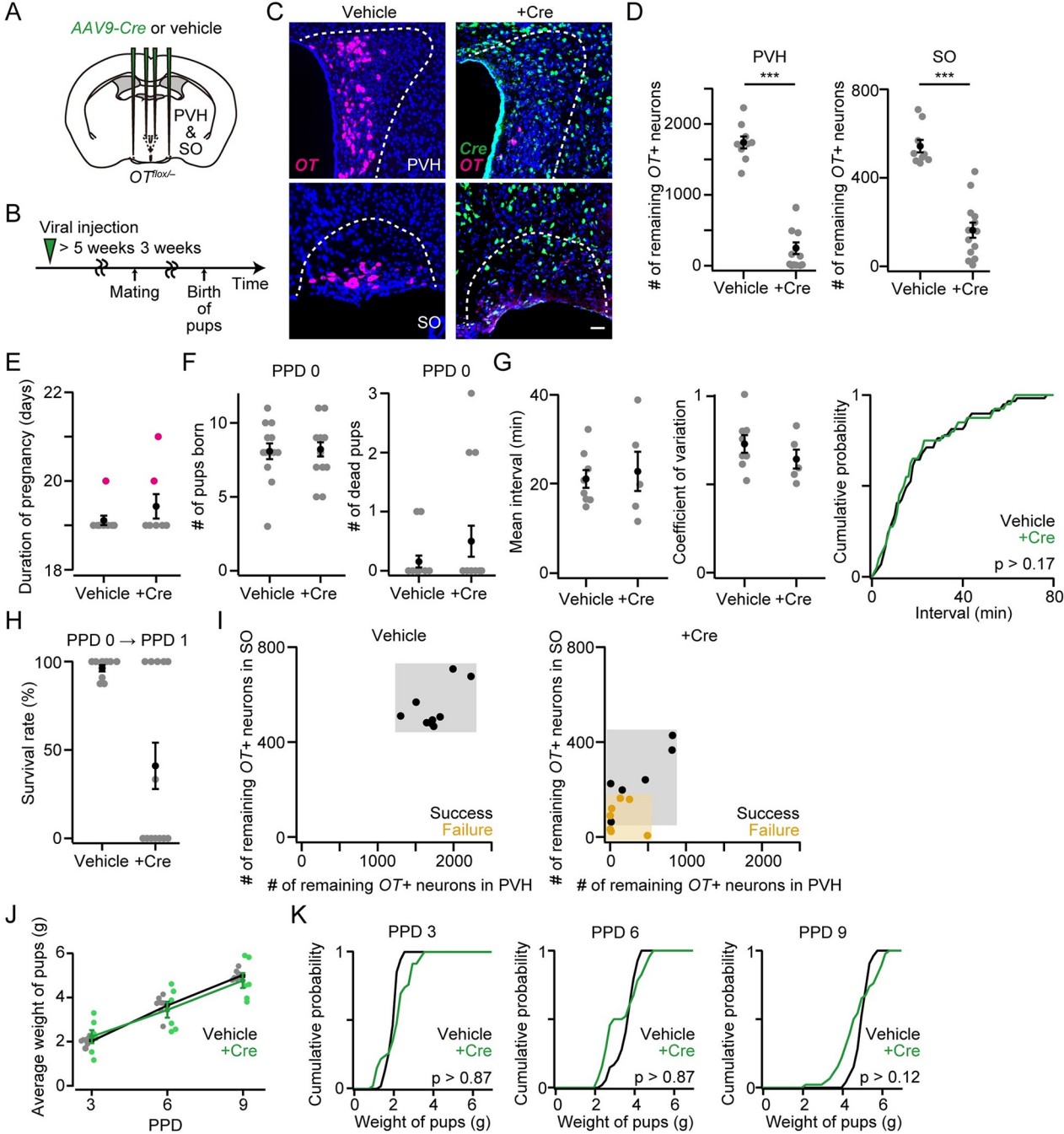

**Fig 1. Grossly normal parturition and abnormal pup survival in *OT* cKO mothers.** (A) Schematic of the viral injection. *AAV-Cre* or vehicle was injected into both the bilateral PVH and SO. (B) Schematic of the timeline of the experiment. (C) Representative coronal sections. *OT* and *Cre in situ* staining are shown in magenta and green, respectively. Blue, DAPI. Scale bar, 20 μm. (D) Number of remaining *OT*+ neurons in the PVH (left) or SO (right). ***p < 0.001, two-sided Mann–Whitney *U*-test. n = 9 and 13 mothers for vehicle and +Cre, respectively. (E) The duration of pregnancy was not statistically different (p > 0.56, two-sided Mann–Whitney *U*-test). n = 9 and 7 for vehicle and +Cre, respectively. Magenta dots indicate stillbirth. (F) Left, number of pups born from the vehicle or +Cre mothers was not statistically different (p > 0.90, two-sided Mann–Whitney *U*-test). Right, number of pups that died within the day of delivery (PPD 0). n = 13 and 14 mothers for vehicle and +Cre, respectively. (G) Left, mean pup delivery interval in each dam. Middle, coefficient of variation showing the interval variability. Right, cumulative probability of interval. n = 59 and 40 pups for vehicle and +Cre, respectively. The p-value is shown in the panel (Kolmogorov–Smirnov test). n = 8 and 5 mothers for vehicle and +Cre, respectively. (H) Survival rate of pups at PPD 1. Note that pups that died at PPD 0 (F) were excluded. n = 9 and 13 mothers for vehicle and +Cre, respectively. (I) Relationship between the success (black dots) or failure (orange dots) of raising pups at PPD 1 and the number of remaining *OT*+ neurons in the PVH (x-axis) or SO (y-axis). Data from the same mice shown in D. (J) Average weight of pups per dam. Mothers in which all pups were dead at PPD 1 were excluded from +Cre. No statistical difference was found in vehicle and +Cre (two-way ANOVA with repeated measurements). n = 7 and 6 for

vehicle and +Cre, respectively. (K) Cumulative probability of weight of pups at PPD 3, PPD 6, and PPD 9. Mothers in which all pups were dead at PPD 1 were not included in +Cre. The p-value is shown in the panel (Kolmogorov–Smirnov test). n = 53–54 and 46 pups from seven and six mothers for vehicle and +Cre, respectively. Error bars, standard error of the mean.

found that the smaller the number of remaining *OT+* neurons in PVH and SO, the more likely the failure phenotype appeared (Fig 1I). In particular, in this mouse line, when the remaining *OT+* neurons in the SO were fewer than 200 (orange shadow in Fig 1I), mothers mostly failed to raise their pups at PPD 1. Together with the observation that *OT* KO dams failed to eject milk [4–6], the failure of raising pups associated with *OT* cKO dams was most likely due to defects in milk ejection. These results support the indispensable functions of OT ligands secreted from the PVH and/or SO for milk ejection as essential for raising pups.

If pups born to *OT* cKO mothers survive PPD 1, do they develop normally or show any growth defects? Given that several dams that had received *AAV-Cre* in both PVH and SO successfully raised pups after PPD 1, we measured the weight of those pups at PPD 3, PPD 6, and PPD 9. We found that the average pup weights of each dam were not statistically different from those of the controls (Fig 1J). Also, the weight distribution of the pups was not statistically different from that of the pups born to vehicle-injected control mothers (Fig 1K). The development of pups was weakly correlated with the number of remaining *OT+* neurons (S1 Fig). These results indicate that the growth of pups is normal even under *OT* cKO in PVH and SO as long as pups survive PPD 1, suggesting that the reduced OT release in cKO mothers affects the onset of lactation most severely.

## Loss of *OT* gene from PVH and SO minimally affects maternal behaviors

OT has been shown to promote maternal caregiving behaviors in rodents [18, 19]. However, previous studies with *OT* KO mothers have reported normal pup-directed parental behaviors [4, 5, 7], except under food-limited stressful environments [8]. To examine whether *OT* cKO mothers exhibit any defects in maternal caregiving behaviors, we analyzed the *OT* cKO mothers characterized in Fig 1 (see Materials and Methods). After injecting *AAV-Cre* into the bilateral PVH and SO, a behavioral assay was performed at PPD 3 (Fig 2A and 2B). We found that all mothers showed retrieval, irrespective of the *Cre* expression or success/failure phenotypes of raising pups (Fig 2C). The latency to investigate the pups, number of retrieved pups, and parental care duration were not significantly different (Fig 2D–2F), suggesting that parental behaviors were unaffected by *OT* cKO in the PVH and SO. A significant difference was found in the time course of retrieval, as shown in the cumulative probability of retrieval (Fig 2G): mothers that received *AAV-Cre* required more time to retrieve pups. Among *OT* cKO mothers, those who exhibited the success phenotype of raising pups at PPD 1 were slightly but significantly slower in pup retrieval compared with control mothers, whereas those who showed the failure phenotype were much slower (Fig 2H). As the *OT* cKO mothers who succeeded in raising pups had more abundant experiences of caring for pups, the observed difference in retrieval latency may be explained by the amount of maternal learning. Taken together, although slight defects could be observed in the efficiency of pup retrieval, the overall performance of parental behaviors in mothers was unaffected by *OT* cKO in the PVH and SO.

## Virgin females with a loss of *OT* gene from PVH and SO were more likely to ignore pups

Although OT neurons are not necessary to induce caregiving behaviors in *OT* KO mothers [5, 7] and mothers of *OT* cKO in the PVH and SO (Fig 2), significant defects in the expression of

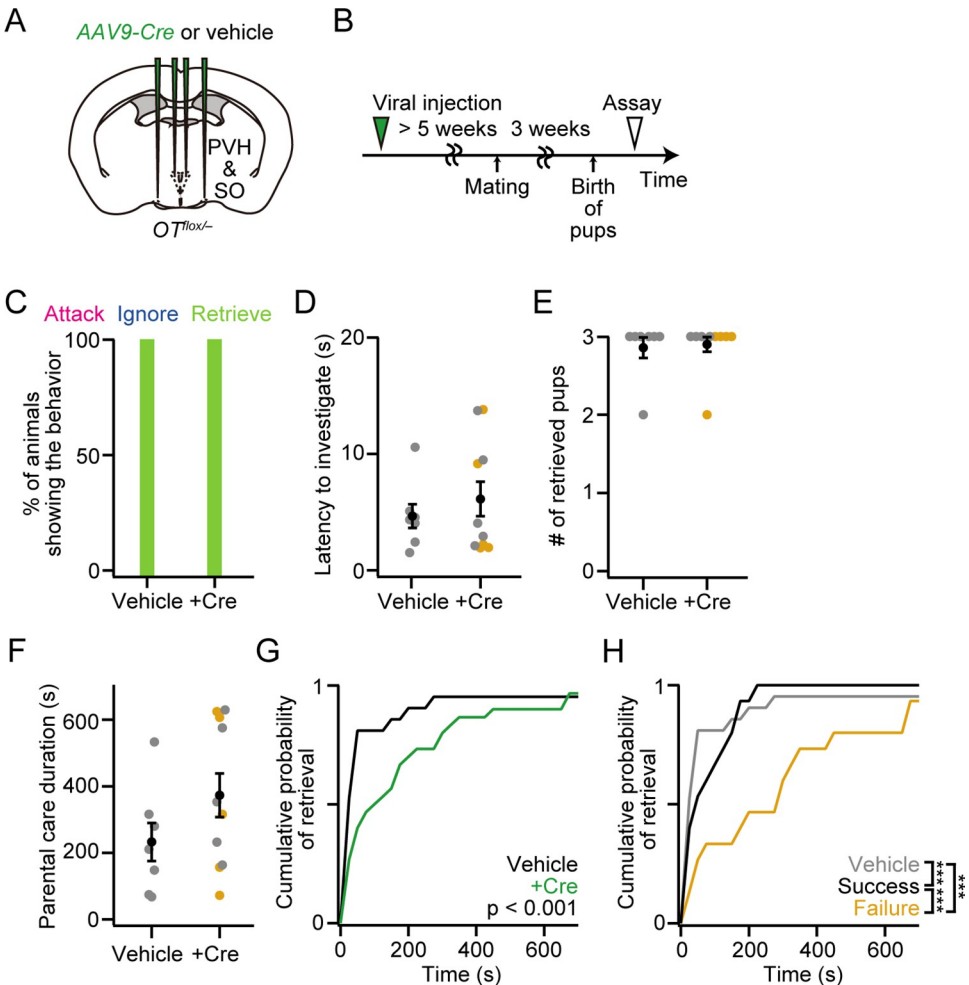

**Fig 2. Grossly normal parental behaviors of mothers with *OT* cKO in the PVH and SO.** (A) Schematic of the viral injection. *AAV-Cre* or vehicle was injected into the bilateral PVH and SO. (B) Schematic of the timeline of the experiment. A behavioral assay was conducted at PPD 3. (C) Percentage of mothers showing attack, ignore or retrieve. Note that attack or ignore was not observed in our dataset. (D) Latency to the first investigation of pups was not statistically different (two-sided Mann–Whitney *U*-test). (E) Number of retrieved pups. (F) Parental care duration was not statistically different (two-sided Mann–Whitney *U*-test). (G) Cumulative probability of pup retrieval. The p-value is shown in the panel (Kolmogorov–Smirnov test). (H) Cumulative probability of pup retrieval of vehicle dams (gray line), and +Cre dams that exhibited success (black line) or failure (orange line) in raising pups at PPD 1 (***p < 0.001, Kolmogorov–Smirnov test with Bonferroni correction). n = 5 each for the success and failure phenotypes, respectively. n = 7 and 10 for vehicle and +Cre from the mice analyzed in Fig 1D, respectively. Orange dots indicate mothers with the failure phenotype. Error bars, standard error of the mean.

maternal caregiving behaviors were observed in *OT* KO virgin females [20]. Furthermore, a recent study reported that virgin females co-housed with experienced mothers and pups began to show caregiving behaviors, and chemogenetic inhibition of OT neurons in the PVH impaired the expression of such alloparental caregiving behaviors [21]. Therefore, we next tested if *OT* cKO induces any defects in the expression of caregiving behaviors in virgin females. We performed cKO of *OT* gene from bilateral PVH and SO in virgin females (Fig 3A). Those females were allowed to co-house with a dam and pups for six consecutive days (Fig 3B; Materials and Methods). We confirmed a drastic reduction of the number of *OT+* neurons in the PVH and SO (Fig 3C). Consistent with the previous study [21], virgin females that received *AAV-Cre* injection showed a significantly higher rate of ignore (Fig 3D;

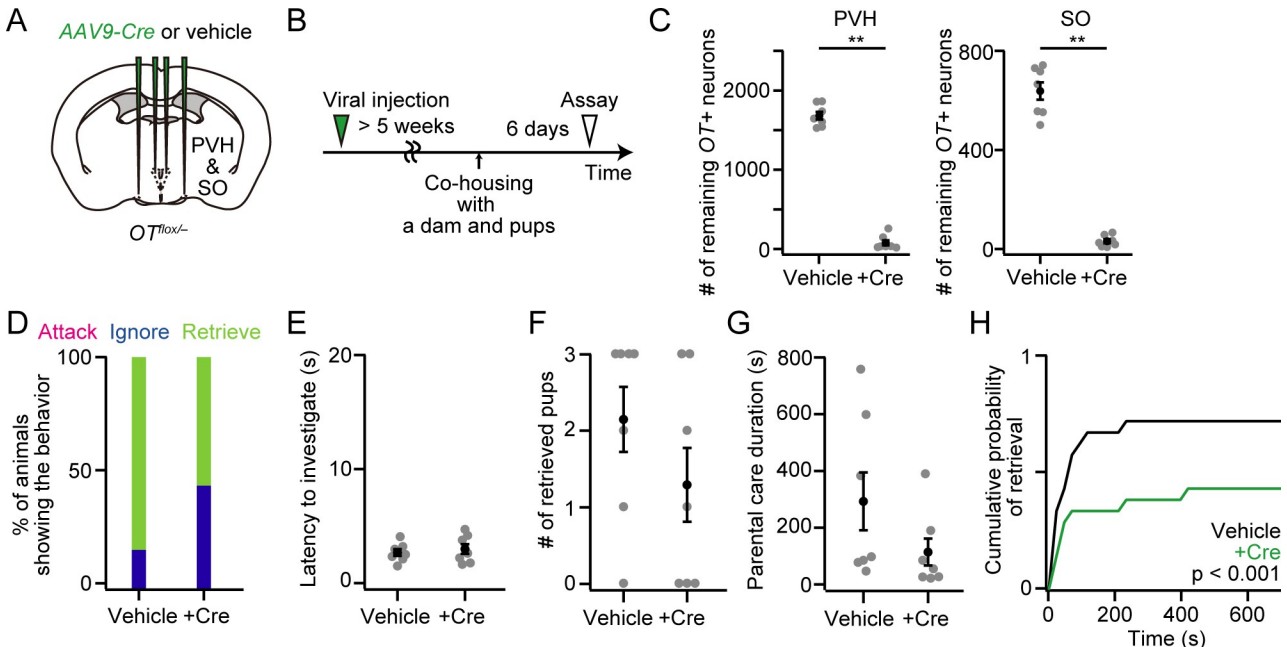

**Fig 3. *OT* cKO in the PVH and SO in virgin females increases the probability to ignore pups.** (A) Schematic of the viral injection. *AAV-Cre* or vehicle was injected into the bilateral PVH and SO. (B) Schematic of the timeline of the experiment. A behavioral assay was performed 6 days after co-habiting with a mother and pups (see Materials and Methods). (C) Number of remaining *OT*+ neurons (**p < 0.01, two-sided Mann–Whitney *U*-test). n = 7 each. (D) Percentage of females showing attack, ignore or retrieve. +Cre females were more likely to ignore the pups (p < 0.001, two-tailed Fisher's exact test). (E) Latency to the first investigation of pups was not statistically different (two-sided Mann–Whitney *U*-test). (F) Number of retrieved pups (p > 0.27, two-sided Mann–Whitney *U*-test). (G) A decrease in parental care duration was found in +Cre females but did not reach the level of statistical significance (p > 0.15, two-sided Mann–Whitney *U*-test). (H) Cumulative probability of pup retrieval. The p-value is shown in the panel (Kolmogorov–Smirnov test). Error bars, standard error of the mean.

p < 0.001, two-tailed Fisher's exact test). While latency to investigate did not differ (Fig 3E), decreases in the number of retrieved pups (Fig 3F) and parental care duration (Fig 3G) were found in +Cre females but did not reach the level of statistical significance (p > 0.27 and 0.15 for Fig 3F and 3G, respectively, two-sided Mann–Whitney *U*-test). Similar to the mothers (Fig 2G), virgin females that received *AAV-Cre* required more time to retrieve pups (Fig 3H). Taken together, these results suggest that loss of *OT* expression leads to a modest but significant defect in the expression of alloparental caregiving behaviors in virgin females.

## Cell ablation of OT neurons in both the PVH and SO recapitulates the *OT* cKO phenotypes

OT neurons release not only OT, but also other neurotransmitters or neuropeptides, such as glutamate [15]. If such non-OT ligands released from OT neurons have any additional functions over OT in labor, milk ejection, or the raising of pups, cell-type-specific ablation of OT neurons would exhibit more severe phenotypes than would those observed in *OT* cKO dams. To examine this possibility, we performed cell ablation of OT neurons by injecting taCasp3-encoding AAV [22] into both the PVH and SO of *OT*^Cre/+ mice (Fig 4A and 4B). As a result, the taCasp3-encoding AAV reduced the number of neurons expressing *OT* mRNA in the PVH and SO (Fig 4C and 4D) most likely because the virus induced cell death in OT neurons [22]. The females became pregnant, and the number of littered pups was not statistically different from that of the controls (Fig 4E). Similar to the *OT* cKO (Fig 1), we found that 33% (= 3/9) of the +taCasp3 dams failed to raise their pups at PPD 1 (Fig 4F). The dams with a smaller

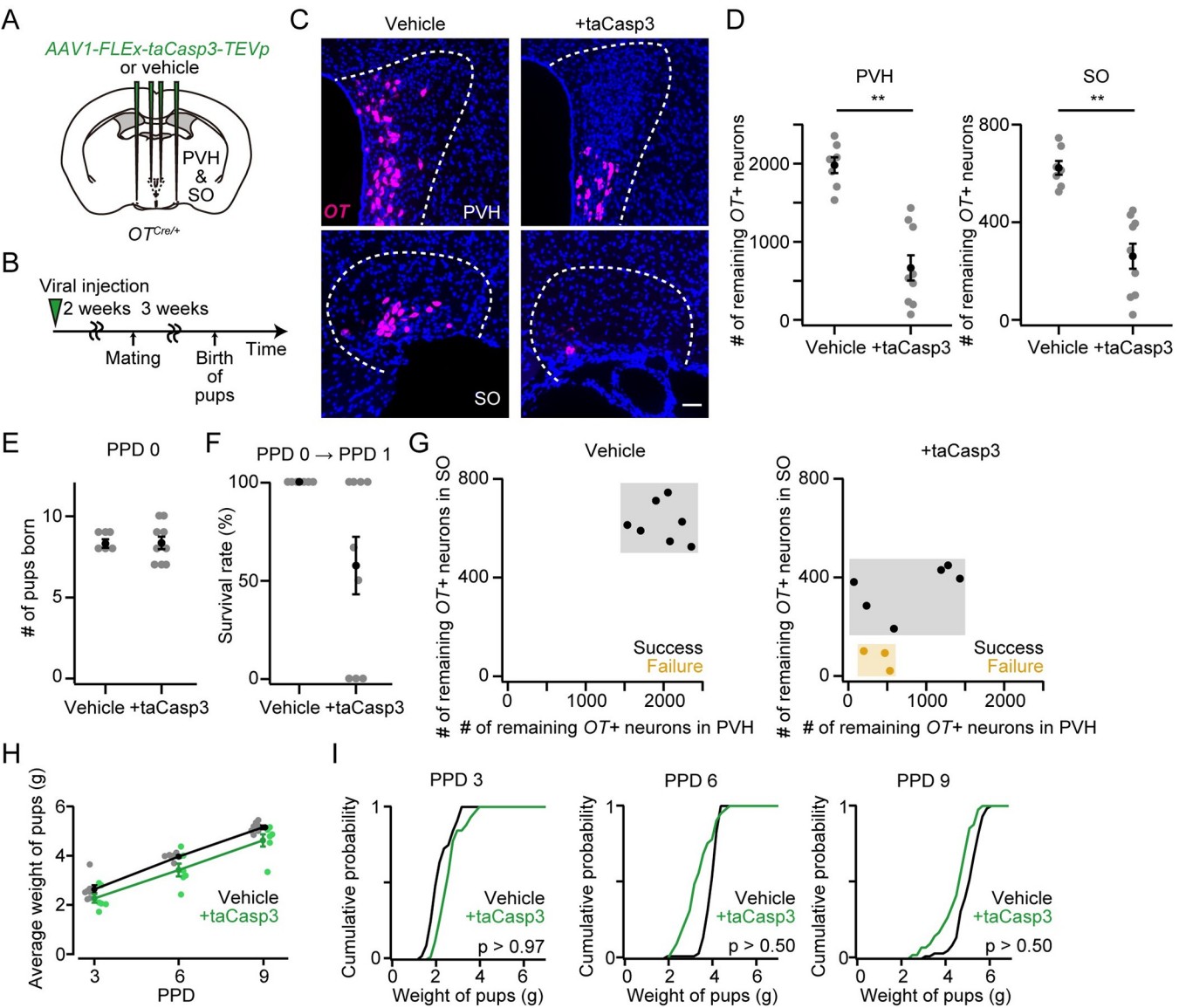

**Fig 4. Cell ablation of OT neurons in the PVH and SO leads to a failure of raising pups.** (A) Schematic of the viral injection. *AAV-FLEx-taCasp3-TEVp* or vehicle was injected into the bilateral PVH and SO. (B) Schematic of the timeline of the experiment. (C) Representative coronal sections. *OT in situ* staining is shown in magenta. Blue, DAPI. Scale bar, 20 μm. (D) Number of remaining *OT*+ neurons in the PVH (left) or SO (right). **p < 0.01, two-sided Mann–Whitney *U*-test. n = 7 and 9 mothers for vehicle and +taCasp3, respectively. (E) The number of pups born was not statistically different (two-sided Mann–Whitney *U*-test). No dead pups were found. n = 7 and 9 mothers for vehicle and +taCasp3, respectively. (F) Survival rate of pups at PPD 1. n = 7 and 9 mothers for vehicle and +taCasp3, respectively. (G) Relationship between the success (black dots) or failure (orange dots) of raising pups at PPD 1 and the number of remaining *OT*+ neurons in PVH (x-axis) or SO (y-axis). Data from the same mice shown in D. (H) Average weight of pups per dam. Mothers in which all pups were dead at PPD 1 were excluded from +taCasp3. No statistical difference was found in vehicle and +taCasp3 (two-way ANOVA with repeated measurements). n = 7 and 6 for vehicle and +taCasp3, respectively. (I) Cumulative probability of weight of pups at PPD 3, PPD 6, and PPD 9. Mothers in which all pups were dead at PPD 1 were excluded in +taCasp3. The p-value is shown in the panel (Kolmogorov–Smirnov test). n = 58 and 41–44 pups from seven and six mothers for vehicle and +taCasp3, respectively. Error bars, standard error of the mean.

number of the remaining *OT*+ neurons in the SO were more likely to show the failure phenotype at PPD 1 (Fig 4G). We also found that neither the average pup weights in each dam nor the weight distribution of the pups was statistically different from the controls (Fig 4H and 4I). These results largely recapitulate the phenotypes observed in *OT* cKO in both the PVH and SO

(Fig 1) without additional defects. These data do not suggest an additional role of non-OT ligands expressed in the OT neurons in pregnancy, parturition, or milk ejection.

## Loss of OT neurons from PVH and SO minimally affects maternal behaviors

To examine the functional role of OT neurons, including non-OT neurotransmitters or neuropeptides, we further analyzed the parental caregiving behaviors of mothers with cell ablation of OT neurons (Fig 5A and 5B) characterized in Fig 4. Similar to *OT* cKO in the PVH and SO (Fig 2), we found no major defects in the execution of caregiving behaviors (Fig 5C–5F), except that dams expressing *taCasp3* required more time to retrieve pups (Fig 5G and 5H). Taken together, under the conditions of cell-type-specific ablation experiments, OT neurons were not necessary to execute maternal caregiving behaviors.

## OT ligands from the SO are needed for the success of raising pups

The results of the cKO of the *OT* gene (Fig 1) and cell ablation of OT neurons (Fig 4) together imply that mothers with a smaller number of the remaining *OT+* neurons in the SO are more likely to fail in raising pups at PPD 1 compared to that in the PVH. Given that our approach enables the cKO of the *OT* gene restricted to a single hypothalamic nucleus, first, we performed cKO of the *OT* gene selectively in the PVH (Fig 6A–6C). The number of littered pups was not significantly different (Fig 6D) and all dams successfully raised pups at PPD 1 (Fig 6E and 6F). We found no defects in the development of pups in the PVH-specific *OT* cKO mothers (Fig 6G and 6H).

Next, we performed cKO of the *OT* gene in the SO (Fig 7A–7C). The number of littered pups was not statistically different (Fig 7D). Similar to the *OT* cKO in both the PVH and SO (Fig 1), 50% (= 4/8) of dams showed a failure in raising pups at PPD 1 (Fig 7E). As expected, dams with a smaller number of the remaining *OT+* neurons in the SO, but not in the PVH, were more likely to show the failure phenotype: approximately 200 remaining *OT+* neurons in the SO were needed for success in raising pups at PPD 1 (Fig 7F). These results suggest that *OT+* neurons in the PVH and SO show differential contributions to pup survival. As described above, we measured the weight of pups at PPD 3, PPD 6, and PPD 9 to analyze their growth. Overall, the growth of the pups was not largely different between +Cre and vehicle (Fig 7G and 7H).

## Discussion

Parturition and lactation are complex biological processes and OT has been considered to be critically involved in both [23, 24]. However, previous studies with whole-body KO of *OT* have reported that only milk ejection was severely impaired, whereas parturition was less dependent on *OT* [4–6]. Decades of studies have suggested that animals have mechanisms to maintain fitness in the presence of harmful mutations. Although this can be achieved by multiple strategies, one possible explanation for the dispensability of OT on the parturition can be the existence of compensatory mechanisms: whole-body KO of *OT* may enhance the expression of related gene(s) [9, 10], thereby compensating for the absence of *OT*. The findings of the present study do not support this explanation, given that cKO of the *OT* gene at the adult stage mostly phenocopied the whole-body KO. Furthermore, we showed that cell ablation of OT neurons resulted in similar phenotypes obtained from *OT* cKO. These findings, together with classical KO studies, suggest that OT is facilitatory but dispensable for parturition.

In the present study, we performed cKO of *OT* gene by injecting *AAV-Cre* into the PVH and/or SO. Our approach enables the removal of the *OT* gene restricted to the brain, and even

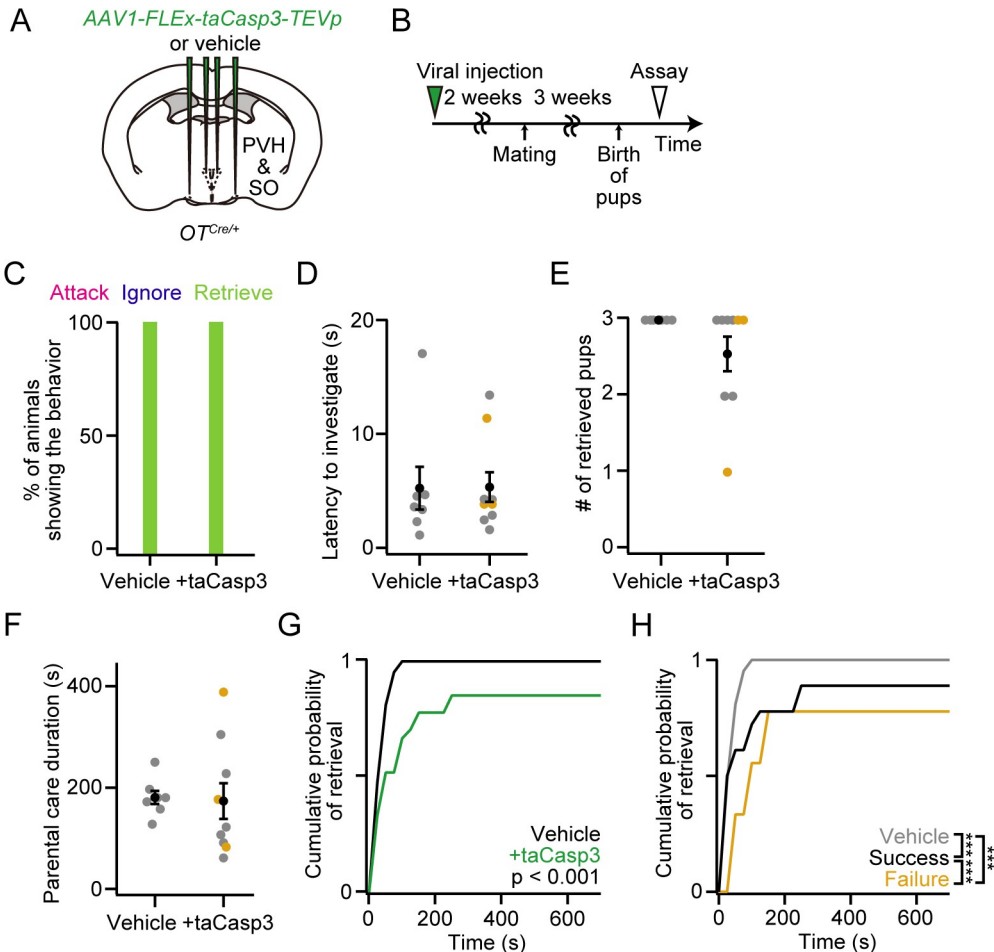

**Fig 5. Grossly normal parental behaviors of mothers with cell ablation of OT neurons.** (A) Schematic of the viral injection. *AAV-FLEx-taCasp3-TEVp* or vehicle was injected into the bilateral PVH and SO. (B) Schematic of the timeline of the experiment. A behavioral assay was conducted at PPD 3. (C) Percentage of mothers showing attack, ignore or retrieve. Note that attack or ignore was not observed in our dataset. (D) Latency to the first investigation of pups was not statistically different (two-sided Mann–Whitney *U*-test). (E) Number of retrieved pups. (F) Parental care duration was not statistically different (two-sided Mann–Whitney *U*-test). (G) Cumulative probability of pup retrieval. The p-value is shown in the panel (Kolmogorov–Smirnov test). (H) Cumulative probability of pup retrieval of vehicle dams (gray line), and +taCasp3 dams that exhibited success (black line) or failure (orange line) in raising pups at PPD 1 (***p < 0.001, Kolmogorov–Smirnov test with Bonferroni correction). n = 6 and 3 for the success and failure phenotypes, respectively. n = 7 and 9 for vehicle and +taCasp3 from the mice analyzed in Fig 4D, respectively. Orange dots indicate mothers with the failure phenotype. Error bars, standard error of the mean.

to a single hypothalamic nucleus, providing a resolution that exceeds previous studies. By visualizing the mRNA of *OT*, we found that the number of remaining *OT+* neurons in SO correlates well with the survival rate of pups (Figs 1I and 7F): the success of raising pups at PPD 1 requires more than 200 *OT+* neurons in the SO, though this number may differ depending on the genetic background. What are the underlying mechanisms? During the milk ejection reflex, the pulsatile OT secretion necessary for the contraction of the mammary glands is mediated by synchronous bursts of OT neurons in the PVH and SO of both hemispheres [25, 26]. In one scenario, there is a pulse generator of synchronous spiking of OT neurons in the SO, the activity of which is then transmitted to all OT neurons, including those in the PVH. This intra- and inter-nucleus transmission of activity may be mediated by OT-to-OTR signaling [27–29]. In this case, loss-of-function of OT neurons in the SO would impair the burst firing

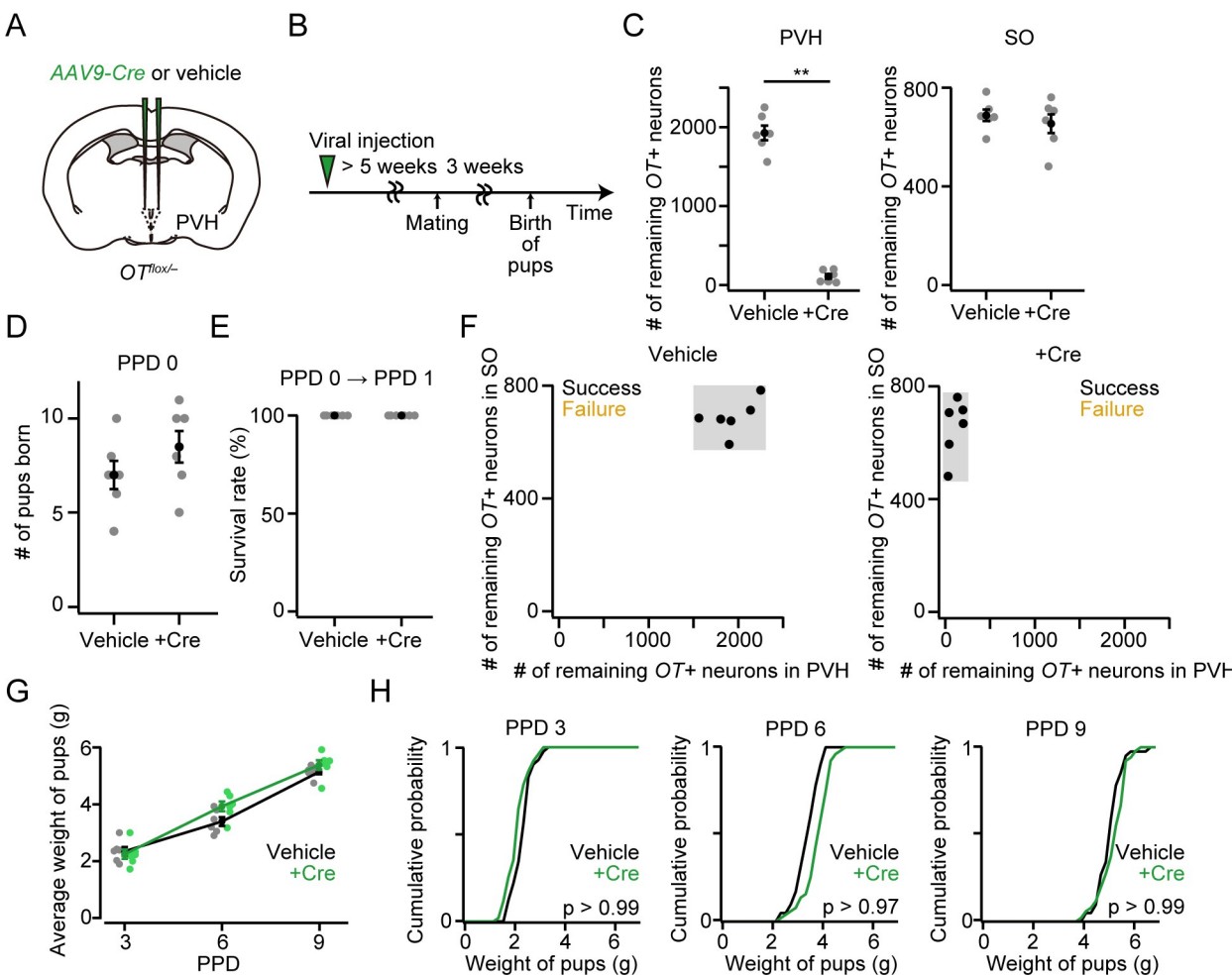

**Fig 6. *OT* expression in the PVH does not affect the survival of pups.** (A) Schematic of the viral injection. *AAV-Cre* or vehicle was injected into the bilateral PVH. (B) Schematic of the timeline of the experiment. (C) Number of remaining *OT*+ neurons in the PVH (left) or SO (right). **p < 0.01, two-sided Mann–Whitney *U*-test. n = 6 and 6 mothers for vehicle and +Cre, respectively. (D) The number of pups born was not statistically different (p > 0.26, two-sided Mann–Whitney *U*-test). No dead pups were found. n = 6 and 6 mothers for vehicle and +Cre, respectively. (E) Survival rate of pups at PPD 1. (F) Relationship between the success (black dots) or failure (orange dots) of raising pups at PPD 1 and the number of remaining *OT*+ neurons in the PVH (x-axis) or SO (y-axis). Data from the same mice shown in C. (G) Average weight of pups per dam. No statistical difference was found in vehicle and +Cre (two-way ANOVA with repeated measurements). n = 6 and 6 for vehicle and +Cre, respectively. (H) Cumulative probability of weight of pups in PPD 3, PPD 6, and PPD 9. The p-value is shown in the panel (Kolmogorov–Smirnov test). n = 41 and 51 pups from 6 and 6 mothers for vehicle and +Cre, respectively. Error bars, standard error of the mean.

of OT neurons in the PVH. Alternatively, the OT neurons in the PVH may remain active even in the absence of OT release from the SO, but the total amount of OT released into the peripheral circulation is not sufficient for milk ejection. In this scenario, pituitary-projecting magnocellular neurons in the SO make a greater contribution to milk ejection [17, 30]. Recent advances in the optical recording of OT neurons during lactation [31, 32] may help explore these possibilities in future studies.

Our data revealed that if the pups survived at PPD 1, their growth was largely normal. This suggests that there is a bottleneck in the establishment of the milk ejection reflex by PPD 1. Fiber photometry-based imaging studies of OT neurons in lactating mice [32] and rats [31] have revealed that the amplitudes of pulsatile $Ca^{2+}$ transients are the lowest at PPD 1 and increased afterward. These findings may explain why milk ejection at PPD 1 is the most

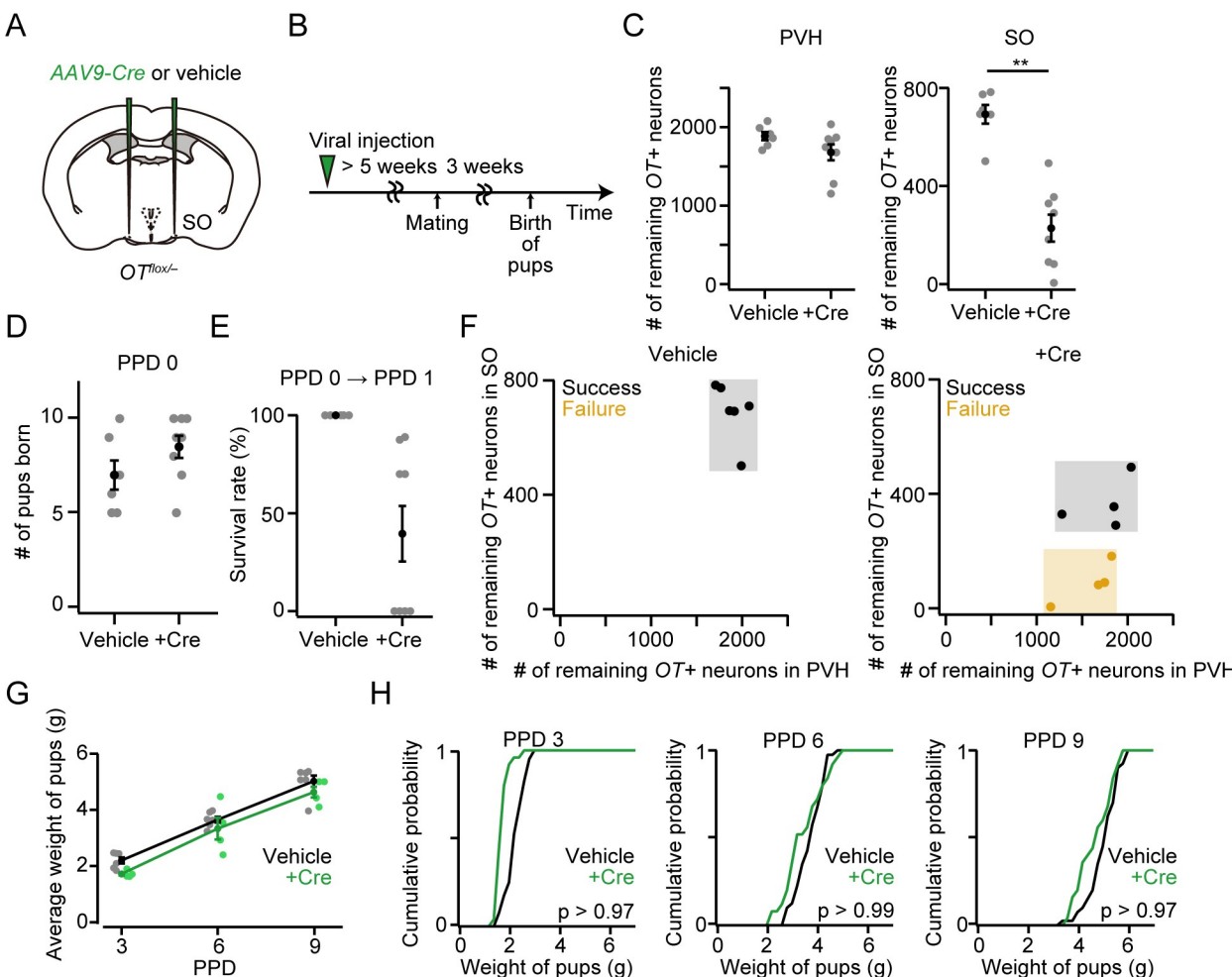

**Fig 7. *OT* expression in the SO is necessary for the survival of pups.** (A) Schematic of the viral injection. *AAV-Cre* or vehicle was injected into the bilateral SO. (B) Schematic of the timeline of the experiment. (C) Number of remaining *OT*+ neurons in the PVH (left) or SO (right). **p < 0.01, two-sided Mann–Whitney *U*-test. n = 6 and 8 mothers for vehicle and +Cre, respectively. (D) The number of pups born was not statistically different (p > 0.21, two-sided Mann–Whitney *U*-test). No dead pups were found. n = 6 and 8 mothers for vehicle and +Cre, respectively. (E) Survival rate of pups at PPD 1. n = 6 and 8 mothers for vehicle and +Cre, respectively. (F) Relationship between the success (black dots) or failure (orange dots) of raising pups at PPD 1 and the number of remaining *OT*+ neurons in the PVH (x-axis) or SO (y-axis). Data from the same mice shown in C. (G) Average weight of pups per dam. Mothers in which all pups were dead at PPD 1 were excluded from +Cre. No statistical difference was found in vehicle and +Cre (two-way ANOVA with repeated measurements). n = 6 and 4 for vehicle and +Cre, respectively. (H) Cumulative probability of weight of pups in PPD 3, PPD 6, and PPD 9. Mothers in which all pups were dead at PPD 1 were excluded in +Cre. The p-value is shown in the panel (Kolmogorov–Smirnov test). n = 40 and 25 pups from six and four mothers for vehicle and +Cre, respectively. Error bars, standard error of the mean.

sensitive to the perturbation of OT neurons: once milk ejection is achieved at PPD 1, the plasticity of OT neurons permits more efficient milk ejection as the pups grow.

In the present study, we showed that *OT* cKO or cell ablation of OT neurons led to delayed pup retrieval in the execution of parental behaviors in mothers, as well as a reduction in the expression of caregiving behaviors in virgin females. In principle, these effects could be due to the experimental procedures employed, such as the injection of AAVs or potential damage to the neural connections caused by the cutting of some afferent or efferent connections of OT neurons. However, negative control experiments that employed the same surgical procedures but just injected a vehicle (saline) showed no phenotype. The likely explanation is that the loss of OT ligands (Fig 2) or OT neurons (Fig 5) may reduce the reward value of pups, thereby

delaying pup retrieval, as the connections from the OT neurons to the dopamine neurons in the ventral tegmental area have been documented in the context of social behaviors [33, 34]. Except for this relatively minor phenotype, all the *OT* cKO or OT neuron-ablated mothers showed normal parental behaviors. Although we cannot fully exclude the possibility that a small number of remaining OT neurons in our virus-based procedures may support these behaviors, the results are in clear contrast to the fathers, given that *OT* KO or *OT* cKO in the PVH lead to defective parental behaviors in fathers [16]. Our observations, together with classical whole-body KO studies, suggest that maternal behaviors are supported by multiple redundant neural systems that can compensate for the loss of OT functions, including the modulation of sensory systems [19, 35]. This likely echoes the evolutionary trait of mammalian species that depends more on maternal care for survival during infancy [36]. The mechanisms of such a redundancy remain an open question. Future studies on how OT facilitates parental behaviors, based on the detailed input–output structures of OT neurons [17, 37], could help define how multiple redundant mechanisms for parental behaviors work in the maternal brain.

## Materials and methods

### Animals

Animals were housed under a 12-hour light/12-hour dark cycle with *ad libitum* access to food and water. Wild-type C57BL/6J mice were purchased from Japan SLC. $OT^{flox/flox}$ and $OT^{-/-}$ mice lines, described previously [16], were generated in C57BL/6J background. We chose the $OT^{flox/-}$ model to increase the efficiency of cKO (Figs 1–3, 6 and 7). If we had used the $OT^{flox/flox}$ mice for cKO, a small fraction of the *flox* alleles that do not experience recombination would easily mask the phenotypes due to the high expression of *OT* gene. $OT^{Cre/+}$ (Jax #024234), purchased from the Jackson Laboratory, was backcrossed more than five generations to C57BL/6J mouse. All the experimental procedures were approved by the Institutional Animal Care and Use Committee of the RIKEN Kobe branch (A2017-15-13).

### Viral preparations

We obtained AAV serotype 9 *hSyn-Cre* from Addgene (#105555, $2.3 \times 10^{13}$ genome particles [gp]/ml). AAV serotype 1 *EF1a-FLEx-taCasp3-TEVp* ($5.8 \times 10^{12}$ gp/ml) [22] was purchased from the University of North Carolina viral core.

### Stereotactic injection

To target AAV into a specific brain region, stereotactic coordinates were defined for each brain region based on the Allen Mouse Brain Atlas [38]. Mice were anesthetized with 65 mg/kg ketamine (Daiichi Sankyo) and 13 mg/kg xylazine (X1251, Sigma-Aldrich) via intraperitoneal injection and head-fixed to stereotactic equipment (Narishige). The following coordinates were used (in mm from the bregma for anteroposterior [AP] and mediolateral [ML], and dorsoventral [DV] from the surface of the brain): PVH, AP –0.8, ML 0.2, DV 4.5; SO, AP –0.7, ML 1.2, DV 5.5. The injected volume of AAV was 200 nl at a speed of 50 nl/min. After the viral injection, the animal was returned to the home cage. Each viral injection and all subsequent experiments except Fig 1E–1G were conducted by two experimenters: experimenter 1 prepared two identical tubes containing either saline (vehicle) or solution containing AAV, and experimenter 2, who was blinded to the contents of the tubes, conducted the injections, behavioral assay, and data analysis.

## Measurement of the duration of pregnancy and pup birth intervals

A virgin female (13 weeks old) individually housed for 5 weeks or longer (Figs 1, 6 and 7) or 2 weeks (Fig 4) after the viral injection was paired with a wild-type male. The next day, the male was removed from the cage and the vaginal plug was checked. Only the females that successfully formed a plug were used for further experiments. On the day of the delivery (PPD 0), typically 12–16 hours after the birth of pups, the number of pups was counted, and dead pups were removed from the cage. In the calculation of the survival rate of pups at PPD 1 (Figs 1H, 4F, 6E, and 7E), the number of dead pups at PPD 0 was excluded. We did not normalize the number of pups, given that the number of pups born was not strongly correlated with the development of pups in our datasets (S1 Fig). As shown in Figs 1I, 2, 4G, 5 and 7, a dam with one or more living pups at PPD 1 was classified as a "success" in raising pups and as a "failure" if all pups were dead at PPD 1. Weight of pups were measured at PPD 3, 6, and 9, in units of 0.1 g. Of note, the genotype of pups born from $OT^{flox/-}$ mothers crossed with wild-type males should be either $OT^{+/-}$ or $OT^{flox/+}$. We found that the ratio of $OT^{+/-}$ pups and $OT^{flox/+}$ pups largely followed Mendel's law and we did not find a statistical significance between mothers that received vehicle or $AAV$-$Cre$ injection to the bilateral PVH (The ratio of $OT^{+/-}$ pups was 62.4 ± 10.4% and 53.1 ± 8.7% in vehicle and +Cre, respectively. $p > 0.81$, two-sided Mann–Whitney $U$-test. n = 6 mothers each). This observation suggests that the genotype of pups did not significantly influence our results.

To measure the pup birth intervals, 18.5 days after the formation of a vaginal plug, each pregnant female was moved to a transparent cage that contained a minimal amount of wood chips and shredded paper, with which the dam built its nest. Two cameras (Qwatch, I-O Data and DMK33UX273, The Imaging Source) were equipped under and on the side of the cage, respectively, to capture the entire cage, including the nest. Videotaping (15 frames/second for Qwatch and 4 frames/second for DMK33UX273) was started after adaptation (5 hours or longer). We defined the birth of each pup as the complete exposure of its entire body from the maternal vagina. In the calculation of the distribution of pup birth intervals, we excluded data from one pup with an interval of longer than 10 hours. Meanwhile, the dam moved freely in the cage, including foraging.

## Parental behavioral assay

**Assay for mothers.** A behavioral assay with mothers was conducted using a similar procedure as described previously [16]. In brief, a virgin female (13 weeks old) individually housed for 5 weeks or longer (Fig 2) or 2 weeks (Fig 5) after the viral injection was paired with a male. The next day, the male was removed from the cage and the vaginal plug was checked. Only the females that successfully formed a plug were used for further experiments. A behavioral assay for the mothers was conducted at PPD 3. For the mothers that successfully fed their pups, all pups were removed from the home cage 6–8 hours before the assay, leaving only the mothers. Unfamiliar wild-type pups (pups unrelated to the resident mother) were used for the assay. Although we prepared three behavioral categories ("Attack", "Ignore", and "Retrieve") as defined previously [16], in our dataset, all dams showed "Retrieve". The following behaviors were further scored: latency to investigate (time after the introduction of pups to the first investigation), pup retrieval, grooming, and crouching. Even if a dam exhibited grooming behavior during crouching, it was only measured as crouching. The duration of animals undergoing either grooming, crouching, or retrieving was scored as parental care duration.

**Assay for virgin females.** Virgin females (Fig 3) were prepared using a similar procedure as described previously [21]. A virgin female (13 weeks old) individually housed for 5 weeks after the viral injection was moved to the cage that contains a resident wild-type mother

rearing the PPD 1 pups. The virgin female was allowed to co-house for 6 consecutive days. One day before the assay, the virgin female was moved to a new cage. The behavioral assay with the virgin females was the same as that with mothers described above.

**In situ hybridization.** Mice were anesthetized with isoflurane and perfused with phosphate-buffered saline (PBS) followed by 4% paraformaldehyde (PFA) in PBS. The brain was post-fixed with 4% PFA overnight. Twenty-micron coronal brain sections were obtained from the entire PVH and SO (typically 32 sections per brain) using a cryostat (Leica), and all sections were subjected to the staining and cell counting. Fluorescent *in situ* hybridization was performed as previously described [16, 39]. In brief, sections were treated with TSA-plus Cyanine 3 (NEL744001KT, Akoya Biosciences) or TSA-plus biotin (NEL749A001KT, Akoya Biosciences) followed by streptavidin-Alexa Fluor 488 (S32354, Invitrogen). The primers (5'– 3') to produce RNA probes were: *OT*, forward, `AAGGTCGGTCTGGGCCGGAGA`, reverse, `TAAGC-CAAGCAGGCAGCAAGC`, *Cre*, forward, `CCAAGAAGAAGAGGAAGGTGTC`, reverse, `ATCCCCA-GAAATGCCAGATTAC` [16]. Brain images were acquired using an Olympus BX53 microscope equipped with a 10× (N.A. 0.4) objective lens. Cells were counted manually using the ImageJ Cell Counter plugin.

## Data analysis

All mean values are reported as mean ± standard error of the mean. The statistical details of each experiment, including the statistical tests used, the exact value of n, and what n represents, are shown in each figure legend. The p-values are shown in each figure legend or panel; non-significant values are not noted.

## Supporting information

**S1 Fig.** (A) Relationship between the average weight of pups and the number of remaining *OT*+ neurons in the PVH and SO (left), PVH (middle), and SO (right). Data was obtained from six $OT^{flox/-}$ mice that received *AAV-Cre* injection into the bilateral PVH and SO. Left, $R^2$ = 0.28, 0.10, and 0.38 for PPD 3, PPD 6, and PPD 9, respectively, p > 0.19 for all PPDs. Middle, $R^2$ = 0.27, 0.07, and 0.37 for PPD 3, PPD 6, and PPD 9, respectively, p > 0.20 for all PPDs. Right, $R^2$ = 0.29, 0.18, and 0.37 for PPD 3, PPD 6, and PPD 9, respectively, p > 0.19 for all PPDs. (B) In our datasets, the number of pups born was not strongly correlated with the development of pups ($R^2$ = 0.01, 0.10, and 0.10 for PPD 3, PPD 6, and PPD 9, respectively. p > 0.47 for all PPDs). Data were obtained from seven $OT^{flox/-}$ mice that received vehicle injection into the bilateral PVH and SO. Note that data points from two mothers of 8 litters nearly overlap. (TIF)

## Acknowledgments

We wish to thank Satsuki Irie for her technical support. We also thank Kumi Kuroda and the members of the Miyamichi Lab for critical reading of the manuscript.

## Author Contributions

**Conceptualization:** Kazunari Miyamichi, Kengo Inada.

**Funding acquisition:** Kazunari Miyamichi, Kengo Inada.

**Investigation:** Mitsue Hagihara, Kengo Inada.

**Methodology:** Mitsue Hagihara.

**Supervision:** Kazunari Miyamichi.

**Writing – original draft:** Kazunari Miyamichi, Kengo Inada.

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
