## [Decision Letter · Decision Letter 0]

8 Nov 2022

PONE-D-22-26274Dispensability of Oxytocin for Parturition and Maternal Behaviors by Conditional Knockout MicePLOS ONE

Dear Dr. Miyamichi,

Thank you for submitting your manuscript to PLOS ONE. After careful consideration, we feel that it has merit but does not fully meet PLOS ONE’s publication criteria as it currently stands. Therefore, we invite you to submit a revised version of the manuscript that comprehensively addresses the points raised during the review process.

We look forward to receiving your revised manuscript.

Kind regards,

Michael Schubert

Academic Editor

PLOS ONE

Journal Requirements:

"K.I. was supported by the RIKEN Special Postdoctoral Researchers Program (https://www.riken.jp/en/careers/programs/spdr/),  a grant from the Kao Foundation for Arts and Sciences (https://www.kao-foundation.or.jp/english.html), and Japan society promotion science KAKENHI (19J00403 and 19K16303) (https://www.jsps.go.jp/english/)

K.M. was supported by Japan society promotion science  (20K20589 and 21H02587) (https://www.jsps.go.jp/english/).

We note that one or more of the authors is affiliated with the funding organization, indicating the funder may have had some role in the design, data collection, analysis or preparation of your manuscript for publication; in other words, the funder played an indirect role through the participation of the co-authors. If the funding organization did not play a role in the study design, data collection and analysis, decision to publish, or preparation of the manuscript and only provided financial support in the form of authors' salaries and/or research materials, please do the following:

a. Review your statements relating to the author contributions, and ensure you have specifically and accurately indicated the role(s) that these authors had in your study. These amendments should be made in the online form.

b. Confirm in your cover letter that you agree with the following statement, and we will change the online submission form on your behalf: 

“The funder provided support in the form of salaries for authors [insert relevant initials], but did not have any additional role in the study design, data collection and analysis, decision to publish, or preparation of the manuscript. The specific roles of these authors are articulated in the ‘author contributions’ section.

Reviewers' comments:

Reviewer's Responses to Questions

**Comments to the Author**

1. Is the manuscript technically sound, and do the data support the conclusions?

Reviewer #1: Partly

Reviewer #2: Partly

2. Has the statistical analysis been performed appropriately and rigorously? 

Reviewer #1: Yes

Reviewer #2: No

3. Have the authors made all data underlying the findings in their manuscript fully available?

Reviewer #1: Yes

Reviewer #2: Yes

4. Is the manuscript presented in an intelligible fashion and written in standard English?

Reviewer #1: Yes

Reviewer #2: Yes

5. Review Comments to the Author

Reviewer #1: The paper by Hagihara et al. reports interesting new findings on the role of oxytocin in regulating parturition, breastfeeding and maternal behavior in female mice. The Authors are leaders in this field of research and have published outstanding papers on oxytocin and reproduction. Here, by selectively deleting oxytocin, or oxytocinergic neurons, in hypothalamic PVN and/or SO nuclei, they confirm that oxytocin is dispensable for parturition and maternal care but also show that the milk ejection role of oxytocin is specifically supported by SO oxytocinergic neurons. Collectively, the paper is well conceived and well written, the methods are adequate. However, some points should be tackled by the Authors before recommending publication.

Major point

In my opinion, the major result of the paper, that oxytocin neurons of the hypothalamic SO nucleus are specifically involved in milk ejection, especially at the onset of lactation, should be confirmed by selectively deleting oxytocin in the PVN. In these mice, according to the Authors’ hypothesis, a normal, or almost-normal, breastfeeding behavior should be observed. This is the most outstanding result of the paper, and the title could be changed or implemented

Minor points

Throughout the manuscript the Authors claim to possible compensatory mechanisms (“genetic compensation”, “enhance the expression of related genes” … which genes?), that in fully oxytocin KO mice could hinder some phenotypes. They should be clearer, and specifically explain what they mean.

In this context (Introduction), why “PVH-specific OT cKO mice show hyperphagic obesity” is not clear to the reader if the Authors do not specify that OT also has a role in food intake and energy balance in adult animals.

The wide readership of the Journal may be not so confident with the mouse models created and studied in the present investigation. In addition to this, the Materials and Method section is at the end of the manuscript. So, a few sentences in the Result section, better describing the procedures used for obtaining the selective conditional KO mice, could render the paper easier to read and meaningful.

When discussing the possible effects of oxytocin on dopaminergic neurons of the ventral tegmental area and rewarding mechanisms, but also concerning data obtained in the present paper, the Authors should also take into consideration the possibility that some phenotypes could be due to a some extent to cutting of afferent, or efferent, projections to, or from, the SO and/or PVN nuclei and to the involvement of glial cells.

Reviewer #2: Oxytocin is necessary for milk let-down and facilitates maternal retrieval in rodents in the transition from nulliparous to parental. The main sources of oxytocin are the paraventricular nucleus of the hypothalamus and the supraoptic nucleus of the hypothalamus. In the mid-1990’s the first knockout studies of oxytocin demonstrated an absence of milk-letdown and impairments in social behavior in congenital whole-body knockouts. In the current report, the authors evaluate the site-specific role of oxytocin using conditional genetic strategies in primiparous female mice.

Strengths:

This is an interesting report that contributes to our understanding of different hypothalamic populations in physiology and behavior.

Major weaknesses:

On line 139-140, the authors state: “ These results suggest that OT+ neurons in the PVH and SO show differential contributions to pup survival.” They do not have a full dataset to support this claim. They do show that conditional deletion of OT from the SO is sufficient to recapitulate deletion from whole body (prior literature) or their conditional deletion from SO AND PVH. There is a missing experiment and result: to support their claim as quoted above, they need to show that conditional deletion of OT from the PVH does not recapitulate the phenotype. This is a likely outcome, but not demonstrated conclusively here.

Additionally, the by using primiparous females during retrieval testing and not virgins, the manuscript fails to capture the impact of oxytocin in the transition from inexperienced to experienced maternal retrieval behavior. It is well-established that inexperienced virgins are likely to ignore pups in their first interactions with them. However, once pregnant and delivered, primiparous dams very quickly learn to retrieve pups. Elevated estrogens from pregnancy and enhanced oxytocin seem to facilitate this process. Once maternal, retrieval behavior is established and less vulnerable to oxytocin manipulations. In this report, the authors have chosen to test their hypothesis in females who have gone through pregnancy- females who will already behave maternally and for which oxytocin is less important. Had they really wanted to evaluate a site specific contribution of the SON or PVN OT to pup retrieval, they should have tested their retrieval behavior in virgins. Contextualizing their findings with others’ work in this area should include a discussion of Rich et al (Rich ME, deCárdenas EJ, Lee HJ, Caldwell HK (2014) Impairments in the Initiation of Maternal Behavior in Oxytocin Receptor Knockout Mice. PLOS ONE 9(6): e98839, and Carcea et al (Carcea, I., Caraballo, N.L., Marlin, B.J. et al. Oxytocin neurons enable social transmission of maternal behaviour. Nature 596, 553–557 (2021). https://doi.org/10.1038/s41586-021-03814-7)

Minor weaknesses:

In the abstract, the language ending on line 38 should be clarified: “We found that cell ablation of oxytocin neurons leads to no additional abnormalities over the oxytocin conditional knockout, suggesting that non-oxytocin ligands expressed by oxytocin neurons have negligible functions” on the responses measured in this report. “on the responses measured in this report” should be added as clarifying language.

The presentation of the figures makes it unclear if the animals used in figures 1 and 2 are the same as in figures 4 and 5. If they are the same, it makes more sense to combine the figures. If they are not the same, then it would be important to show validation of cKO for this additional cohort of animals. If the figures get unwieldy with multiple panels, the following current figure panels seem unnecessary: Figure 1K, Figure 2I, Figure 3H, Figure 4C, Figure 5C

More details of methods should be included instead of solely referencing prior papers. For example, for the ISH, probe details should be included.

It is unclear why the OTflox/- was necessary or an improvement over OTflox/flox. The rationale for this choice should be explained.

How were cells counted? How many sections per side per animal?

“In situ hybridization” would be a better section heading in the methods than “histochemistry”

The statistical analysis for pup growth should be a repeated measures ANOVA assuming that the animals are traceable as individuals overtime.

The cKO model uses oxytocin floxed in the context of an Oxt knockout a allele bred with males who are wild type. This breeding strategy would produce wild-type and heterozygous offspring. It is unclear what the impact of an interaction between pup genotype and mom genotype might be. This should be addressed in the discussion.

In figure 2 the authors use oxytocin in situ hybridization to evaluate the effectiveness of the caspase virus which clearly shows a reduction in oxytocin. However, their claim is that the caspase method reduces the oxytocin cell population rather than just the oxytocin mRNA. Ideally, they would show evidence of cell loss rather than just oxytocin mRNA loss. Is there a significant reduction in the DAPI positive cell density in the PVN and SON?

There appears to be a lot of variability in the pup growth across ages in virus treated females. Given that the remaining OT+ neurons are associated with pup survival, it would be interesting to probe if the remaining oxytocin neurons in the PVN or SON of dams predict pup growth for pups that do survive.

The lack of a virus control is unfortunate.

Were the litters normalized to the same size? Litter size impacts pup growth- too few pups and too many pups reduce per pup weight. This should probably be modeled. This is particularly important due to the influence of ventral stimulation (intensity of which is influence by the number of pups) which stimulates the Ferguson reflex for milk let-down. Is it possible that the KO is less able to modulate pup growth based on litter size?

6. PLOS authors have the option to publish the peer review history of their article (what does this mean?). If published, this will include your full peer review and any attached files.

Reviewer #1: No

Reviewer #2: No

---

## [Author Response · Author response to Decision Letter 0]

31 Jan 2023

see our "response to the reviewers" doc.

---

## [Decision Letter · Decision Letter 1]

28 Feb 2023

PONE-D-22-26274R1The Importance of Oxytocin Neurons in the Supraoptic Nucleus for Breastfeeding in MicePLOS ONE

Dear Dr. Miyamichi,

Thank you for submitting your manuscript to PLOS ONE. After careful consideration, we feel that it has merit but does not fully meet PLOS ONE’s publication criteria as it currently stands. Therefore, we invite you to submit a revised version of the manuscript that addresses the point raised during the review process.

We look forward to receiving your revised manuscript.

Kind regards,

Michael Schubert

Academic Editor

PLOS ONE

Journal Requirements:

Reviewers' comments:

Reviewer's Responses to Questions

**Comments to the Author**

1. If the authors have adequately addressed your comments raised in a previous round of review and you feel that this manuscript is now acceptable for publication, you may indicate that here to bypass the “Comments to the Author” section, enter your conflict of interest statement in the “Confidential to Editor” section, and submit your "Accept" recommendation.

Reviewer #1: All comments have been addressed

Reviewer #2: All comments have been addressed

2. Is the manuscript technically sound, and do the data support the conclusions?

Reviewer #1: Yes

Reviewer #2: Yes

3. Has the statistical analysis been performed appropriately and rigorously? 

Reviewer #1: Yes

Reviewer #2: No

4. Have the authors made all data underlying the findings in their manuscript fully available?

Reviewer #1: Yes

Reviewer #2: Yes

5. Is the manuscript presented in an intelligible fashion and written in standard English?

Reviewer #1: Yes

Reviewer #2: Yes

6. Review Comments to the Author

Reviewer #1: (No Response)

Reviewer #2: The authors have adequately addressed the original reviews. In the new Figure 3 F, the distribution of the data indicate the possibility that the data needs a non-parametric mann-whitney U test rather than a parametric t-test.

7. PLOS authors have the option to publish the peer review history of their article (what does this mean?). If published, this will include your full peer review and any attached files.

Reviewer #1: No

Reviewer #2: No

---

## [Author Response · Author response to Decision Letter 1]

1 Mar 2023

Response to Reviewer #2:

We are greatly thankful to this reviewer for a positive evaluation of our revision and for providing invaluable suggestions on the statistical analysis. Following a comprehensive analysis of the data structure, we have made the decision to replace the student t-test with a two-sided Mann-Whitney U-test for the following data sets: Fig. 1D, E, F, Fig. 2D, F, Fig. 3C, E, F, G, Fig. 4D, E, Fig. 5D, F, Fig. 6C, D, Fig. 7C, D. It is our belief that this alteration will lead to an enhancement in the statistical accuracy of our results. Importantly, our original conclusions remain unchanged.

---

## [Editor Report · Decision Letter 2]

3 Mar 2023

The Importance of Oxytocin Neurons in the Supraoptic Nucleus for Breastfeeding in Mice

PONE-D-22-26274R2

Dear Dr. Miyamichi,

We’re pleased to inform you that your manuscript has been judged scientifically suitable for publication and will be formally accepted for publication once it meets all outstanding technical requirements.

Kind regards,

Michael Schubert

Academic Editor

PLOS ONE

---

## [Editor Report · Acceptance letter]

8 Mar 2023

PONE-D-22-26274R2 

The Importance of Oxytocin Neurons in the Supraoptic Nucleus for Breastfeeding in Mice 

Dear Dr. Miyamichi:

I'm pleased to inform you that your manuscript has been deemed suitable for publication in PLOS ONE. Congratulations! Your manuscript is now with our production department. 

Kind regards, 

on behalf of

Dr. Michael Schubert 

Academic Editor

PLOS ONE